# Different Types of Deletions Created by Low-Copy Repeats Sequences Location in 22q11.2 Deletion Syndrome: Genotype–Phenotype Correlation

**DOI:** 10.3390/genes13112083

**Published:** 2022-11-10

**Authors:** Eva-Cristiana Gavril, Roxana Popescu, Irina Nucă, Cristian-Gabriel Ciobanu, Lăcrămioara Ionela Butnariu, Cristina Rusu, Monica-Cristina Pânzaru

**Affiliations:** 1Department of Medical Genetics, Faculty of Medicine, “Grigore T. Popa” University of Medicine and Pharmacy, University Street, No 16, 700115 Iasi, Romania; 2Investigații Medicale Praxis, St. Moara de Vant No 35, 700376 Iasi, Romania; 3Department of Medical Genetics “Saint Mary” Emergency Children’s Hospital, St. Vasile Lupu No 62, 700309 Iasi, Romania

**Keywords:** 22q11.2DS, low-copy repeats sequences, phenotype, MLPA

## Abstract

The most frequent microdeletion, 22q11.2 deletion syndrome (22q11.2DS), has a wide and variable phenotype that causes difficulties in diagnosis. 22q11.2DS is a contiguous gene syndrome, but due to the existence of several low-copy-number repeat sequences (LCR) it displays a high variety of deletion types: typical deletions LCR A–D—the most common (~90%), proximal deletions LCR A–B, central deletions (LCR B, C–D) and distal deletions (LCR D–E, F). Methods: We conducted a retrospective study of 59 22q11.2SD cases, with the aim of highlighting phenotype–genotype correlations. All cases were tested using MLPA combined kits: SALSA MLPA KIT P245 and P250 (MRC Holland). Results: most cases (76%) presented classic deletion LCR A–D with various severity and phenotypic findings. A total of 14 atypical new deletions were identified: 2 proximal deletions LCR A–B, 1 CES (Cat Eye Syndrome region) to LCR B deletion, 4 nested deletions LCR B–D and 1 LCR C–D, 3 LCR A–E deletions, 1 LCR D–E, and 2 small single gene deletions: delDGCR8 and delTOP3B. Conclusions: This study emphasizes the wide phenotypic variety and incomplete penetrance of 22q11.2DS. Our findings contribute to the genotype–phenotype data regarding different types of 22q11.2 deletions and illustrate the usefulness of MLPA combined kits in 22q11.2DS diagnosis.

## 1. Introduction

22q11.2 deletion syndrome (22q11.2DS) is the most common microdeletion. It has an incidence of 1/4000 newborns (latest studies suggest an underdiagnosis of the condition, with a real incidence of 1/1000 newborns) [1,2]. Although it has a wide and variable phenotype that causes difficulties in diagnosis, the syndrome is frequently characterized by palatal defects (in 67%) (including cleft palate, submucosal cleft palate, velopharyngeal insufficiency, bifid uvula, hyper nasal speech, and dysphagia) [3], conotruncal heart defects (in 64% of individuals) [4], characteristic cranio-facial dysmorphism (asymmetric crying facies, hooded eyelids, prominent nasal bridge, bulbous nose, micrognathia, ear abnormalities, craniosynostosis) [5], hypocalcemia (in 50% cases), thymus and parathyroid hypoplasia, mild intellectual disability, psychiatric illness, and hearing impairment [6,7].

Not only is the phenotypic variety very high, but also the variety of deletion types and sizes in this region, due to several low-copy-number repeat sequences (LCR22): LCR A, LCR B, LCR C, LCR D, LCR E and LCR F. Most individuals with 22q11.2DS have a 2.54 Mb heterozygous deletion (~90%) [8] extending from LCR A–D—del(22)(q11.2) chr22:18,912,231–21,465,672 (NCBI Build GRCh38), which comprises approximately 40 genes, including *TBX1*. Haploinsufficiency of the *TBX1* gene (located in the region flanked by LCR22A-B) is particularly responsible for most of the phenotypic findings [9,10]. Approximately 5% of individuals with 22q11.2DS have a 1.5 Mb heterozygous deletion extending from LCR A–B (proximal deletion) [11], 2% have deletion extending from LCR A–C and about 5% have a smaller, atypical (‘nested’) heterozygous deletion that extends from LCR B–D or C–D, del(22)(q11.2) chr22:20,731,986–21,465,672 (NCBI Build GRCh38) also called central deletions [12]. Distal deletions, flanked by LCR D–E, LCR D–F, have been reported less frequently [13].

Genomic testing methods that can determine the copy number of sequences are chromosomal microarray or targeted deletion analysis represented by FISH analysis and multiplex ligation-dependent probe amplification (MLPA). By using MLPA, up to ~95–100% of deletions in this region can be identified [14,15]. Also, it is possible to determine the size of a deletion by identifying the LCR22 flanking it.

We conducted a retrospective study of 22q11.2DS cases, with the aim of highlighting phenotype–genotype correlations, especially in new cases of atypical deletions.

## 2. Materials and Methods

We studied 59 cases diagnosed with 22q11.2DS in Iasi Medical Genetics Center. The patients were clinically evaluated periodically over an average period of 9 years. 

All cases were tested using MLPA combined kits: first, an MLPA screening kit for microdeletions (P245—MRC Holland) and then for confirmation and detection of deletion breakpoints, a follow-up MLPA kit (P250—DiGeorge Kit) containing 48 MLPA probes, 31 of them located in the region 22q11.2- 22q13, and thus able to distinguish the most frequent types of deletion; the rest of the probes evaluated regions relevant for DiGeorge syndrome type II or disorders with phenotypic features of DiGeorge syndrome on chromosomes 4q, 8p, 9q, 10p and 17p. The detailed data of the probes included in the utilized MLPA kits are described on the manufacturer’s website. MLPA analysis was applied according to the manufacturer’s instructions. Deletion sizes were determined using the UCSC genome browser (NCBI36/hg18, http://genome.ucsc.edu/).

The study design was conducted according to a protocol approved by the Ethics Commission of “Grigore T. Popa” University of Medicine and Pharmacy, Iasi (Approval code: 14629). Informed consent was signed by all the patients or their parents or legal guardians. Voluntary free entrance was offered to all the patients included in this study. 

In most cases, the parents were also tested; most patients have de novo deletion, only six cases have inherited the deletion (maternal inheritance was noted in all six cases).

## 3. Results

Most of the cases were recorded from infancy or in the first years of childhood. Regarding the age distribution at the last evaluation, the cases were 11%—6 months old to 1 year old, 23%—2 to 6 years old, 34%—6 to 12 years old, 17%—12 to 18 years old and 17%—over 18 years old.

Of the 59 patients studied, 45 (76%) have a classic deletion from LCR A–D (Figure 1a). A total of 14 atypical deletions were also identified: LCR A–B—2 cases (Figure 1b), CES (Cat Eye syndrome region) to LCR B—1 case (Figure 1c), LCR B–D—4 cases (Figure 1d). LCR C–D—1 case, LCR A-E 3 cases, LCR D–E—1 case, and 2 small single gene deletions: del*DGCR8* and del*TOP3B* (Figure 2).

Genotype–phenotype correlation was performed based on the deletion type—typical as shown in Table 1 and atypical—proximal, central and distal deletions, as shown in Table 2. The results are presented in comparison with the literature data in order to facilitate the discussions.

Out of the total of 59 patients, 41 (71%) presented heart defects: 27 males (23.6%) and 14 females (17.3%). Chi-square statistic with Yates correction is 2.72 and p value is 0.1 and is not statistically significant. 

Typical dysmorphia (hooded eyelids, ear anomalies, prominent nasal bridge, bulbous nose, micrognathia, asymmetric crying facies, craniosynostosis) was noted in 64% of cases, and defects of the palate in 60% of cases, in most cases velopalatine insufficiency (difficult feeding in infants or hyper nasal voice); cleft palate was observed in six patients. A statistically significant difference was not observed between male and female patients with velopalatine defects (Chi-square statistic with Yates correction is 0.0176 and *p* value is 0.084).

Among our cases, 59% presented intellectual disability: 16 mild, 11 medium and 7 severe impairments. Behavioral disorders were also observed: attention deficit in eight cases, two cases with autism, two cases with behavioral aggression, and one case of schizophrenia (Table 1 and Table 2).

Various urogenital defects were likewise identified (23%): hydronephrosis, renal hypoplasia, renal agenesis, polycystic kidney, chronic kidney disease and hypospadias.

We identified four cases with proximal deletion (three de novo), all presenting septal defects (ASD, VSD): three of them have palate defects, and mild or atypical dysmorphic faces. Other cases with proximal deletions reported in the literature have highly variable clinical findings. Usually they include congenital heart defect (conotruncal malformations), developmental delay/intellectual disability (DD/ID), behavioral disorders, palatal abnormalities (velopharyngeal incompetence, cleft palate and bifid uvula), characteristic face, hypocalcemia and immune deficiency. Most proximal deletions are de novo [5,16,17].

Our five cases with central deletion (three de novo) have mild clinical phenotypes. The heart defect is the most important feature, present in four out of five patients, dysmorphic face was noticed in four out of five, one case presented DD/ID, and there was one bilateral renal anomaly. Palatal defects were not observed in this type of deletion. In other studies, central deletions (LCR B–D/C, LCR C–D), were associated with cardiac developmental anomalies like those in LCR A-B or LCR A-D deletions, although with lower frequency [11,18,19]. The variable clinical phenotype includes in many cases dysmorphic face, DD/ID, seizures, growth restriction, behavioral problems, immune deficiency, skeletal anomalies, and genitourinary defects. A total of 60% are de novo deletions [17,19,20,21,22,23,24,25,26].

In the distal deletion we noticed a mild phenotype with palatal defects and no heart defects or ID. In other studies, distal deletions have a highly variable phenotype including prematurity, prenatal and postnatal growth deficit, developmental delay, craniofacial dysmorphism, cardiac, skeletal (mild) and urogenital anomalies [27,28]. Cleft palate and high arched palate were noted in two patients with LCR D–F deletion [29].

We also found two large deletions containing regions A–D and D–E, and one with LCR A–beyond D (up to *HIC2* gene in D–E region). In these cases, we noted that the phenotype was more severe, with moderate/severe ID, renal defects, polydactyly, spastic tetraplegia, cerebral atrophy or spina bifida. Literature data include three cases with LCR B–D+ (deletion between *ZNF74* and *HIC2* probes) and clinical features include: ventricular septal defect, cleft lip, and palate, retrognathia, hyper nasal speech, dysmorphic face, short stature, moderate ID and ADHD [11].

## 4. Discussion

In most of the studied patients (76% cases), classic deletion in the 22q11.2 region, between LCR A–D, was identified. However, atypical deletions were observed in 14 cases. A total of 5% presented proximal deletion, and 8.4% presented central deletion. Also, one distal deletion, two small single gene deletions and three larger deletions at the proximal point in LCR A and distally exceeding LCR D were found. Thus, the percentage distribution of different types of deletions in our study is somewhat modified compared to the data from the literature, where typical deletions represent 85–90% of cases [4,9]. However, the limited sample size of our study (59 individuals) may account for the variations seen.

The latest studies suggest that cardiovascular anomalies are present in 49–83% of people with 22q11.2DS [4,30]; the most common are septal defects (23%), followed by tetralogy of Fallot (18%) [5]. In our patients, the prevalence of cardiac defects was 71.1% and a higher occurrence of Fallot tetralogy was observed—45.2% out of the patients with heart defects. Many individuals had multiple cardiac findings and complex congenital heart defects. This could be explained by the fact that our patients are initially evaluated by a cardiologist. Other cardiovascular anomalies include pulmonary stenosis, truncus arteriosus, hypoplastic left ventricle, single ventricle, aortic coarctation, anomalous origin of the left subclavian artery and vena cava superior anomalies. We also observed that in adults, the cardiac phenotype is milder, without heart defects or with mild cardiac anomalies, which supports the fact that the main cause of death in 22q11.2DS is congenital heart disease (~87% of all deaths in children with 22q11) [5,31,32]. The main candidate gene for heart defects is *TBX1* located in the LCR A–B interval [33]. Other genes in this region that were considered as contributing to the development of cardiac anomalies were: *PRODH*, *DGCR6* and *DGCR8* [16,34]. In a large study of congenital heart defects in 22q11.2DS, comprising 1053 cases, it is suggested that although the LCR A–B region was considered the critical region for the 22q11.2 phenotype and especially for heart defects, the presence of congenital cardiac defects in patients with LCR B–D and LCR C–D deletions (20–30% of cases) highlights the importance of the LCR C–D region in normal cardiac development, and proposes haploinsufficiency of the *CRKL* gene as the main factor involved in conotruncal heart defects [35]. Most of our cases with cardiac defects include this region. Four patients that do not have a deletion in this region had associated heart defects: two Fallot tetralogy, one VSD and ASD. This could be explained by the haploinsufficiency of the *CRLK* gene [36].

Regarding velopalatal defects, in 67% of 22q11.2DS cases palatal defects have been reported, the most frequent abnormality being velopharyngeal incompetence [3]. In a meta-analysis study comparing the prevalence of cardiac malformations and palatal defects with the size of the LCR A–B/LCR A–C/LCR A–D deletion (1514 patients from eight studies), Rozas et al. concluded that there is no association between the size of the deletion and clinical abnormalities investigated, and considered the A–B region (overlapping region) as a critical region, outside of which there may still be important modifier genes involved in phenotypes [37]. In our study, palatal anomalies were detected in 35/59 of our cases (59.3%), being almost constant in deletions that include the LCR A–B region. All cases with central deletions (LCR B–D and LCR C–D) did not show velopharyngeal dysfunction, being in concordance with data from the literature [20] and suggesting the involvement of genes from the A–B region in velopalatine damage. However, the presence of velopalatine insufficiency (dysphagia and nasal voice) was found in the distal deletion (LCR D–E), as well as in the small distal deletion including the *TOP3B* gene. This finding underlines the *TOP3B* gene’s role in the development of palatal defects as well as in cognitive impairment and facial dysmorphism observed by other studies [38]. Four other cases with distal deletions including sequence LCR D–E were reported previously as having cleft lip and palate and cleft palate [39].

Most 22q11.2DS patients have a history of hypotonia in childhood [40], but that was not the case in our study, with hypotonia being noted in only eight cases.

Intellectual disability in 22q11.2DS is usually mild, cases with severe intellectual disability being very rare, and usually reported in familial cases [41]. Although most of the cases followed by us presented mild intellectual impairment, severe DI was reported in seven cases, only two of which had inherited chromosomal abnormality [42]. Moderate and severe intellectual disability was observed in cases with distal deletions, highlighting the involvement of the *TOP3B* gene. On the other hand, the relatively high number of cases with severe and moderate intellectual disability in typical deletions can also be attributed to the patients’ socio-economic level, many of them coming from rural areas or families with low income, and low access to proper education.

Common behavioral illnesses include attention deficit disorder, anxiety, and problems in social interactions. According to reports, 20% of individuals with 22q11.2DS have autism and autistic spectrum disorders [4]. A total of 18.6% of the analyzed patients presented attention deficit and behavior disorders and in two cases autistic spectrum disorder was reported; only one case out of the eight adults was diagnosed with schizophrenia. In other studies, 60% of adults with 22q11.2DS develop a psychiatric disorder, especially schizophrenia (25%) [41,43]. The lower frequency of behavioral and psychiatric disorders may also be due to the younger age of many of our patients, with our findings agreeing with data from studies with younger cohorts [44]. In a meta-analysis study on the incidence of psychiatric disorders in 22q11.2DS patients, 1/10 cases with 22q11.2DS have psychotic presentations at diagnosis and approximately 1/10 of the rest will develop psychosis later, cohorts with an average age above 18 years having a higher prevalence [45]. The LCR B–D region was designated as responsible for the appearance of psychological manifestations, especially the *PI4KA* gene [46]. From our affected cases, only two cases with ADHD have a proximal deletion, and therefore did not have haploinsufficiency of this region.

Renal anomalies were present in 23.7% of our patients (14 cases), a slightly higher percentage than in the literature [4,47]; three cases of cryptorchidism and one of hypospadias were also identified. It is considered that boys are more likely to present a genitourinary anomaly than girls, which was not true in our cases, genitourinary anomaly being observed in an almost equal percentage in both sexes (10 boys/8 girls). Different studies suggest that genes from the LCR C–D region are responsible for the occurrence of genitourinary anomalies, the *CRKL* and *LTZR1* genes being the main candidates [5,48]. Only two cases with genitourinary malformation do not cover this region: one case with proximal A–B deletion and one case with a small deletion including only the *DGCR8* gene. The *DGCR8* gene, however, is considered to play an epigenetic role in the symptoms associated with 22q11.2DS, including neuropsychiatric, genitourinary, and other abnormalities, and as a result, may be accountable for both cases [10,49]. 

Regarding immune deficiency, present in the literature in 67% of cases [50,51], in our study it was observed in 35.59% of cases, all with haploinsufficiency of LCR A–B [13].

Hearing impairment was reported in 6–60.3% of 22q11.2DS cases [52]. A total of 10 of our patients (16.9%) were diagnosed with hearing impairment with or without chronic otitis media. The patients with hearing impairment had the length and localization of 22q11.2 deletion of different types: CES to B, A to B, A to D, Del *DGCR8*. Three cases also presented ear anomalies: malformations of the auditory canal preauricular tag or pit and bifid lobe.

Other anomalies found in our studied cases, but rarely described in the literature are: three cases of polydactyly (one extra finger), progressive polyarthritis, two cases with hepatic cytolysis, one case of stroke, two cases of cerebral atrophy and one case of spina bifida.

Although 22q11.2DS is a contiguous gene syndrome, the size of the deletion does not influence the severity of the phenotypic manifestations: LCR A–D deletions most often present similar phenotype and severity compared to smaller proximal deletion (LCR A–B) [53,54]. However, it was observed that central (LCR B–D) and distal deletions (LCR D–E/F) usually have a milder phenotype [55], a fact also observed in our study; at the same time, the incomplete penetrance and variable expressivity can also be noticed, especially in inherited deletions, where the same size and type of deletion can determine a severe phenotype but also a discrete or even asymptomatic one.

## 5. Conclusions

This study emphasizes the wide phenotypic variety from one case to another and incomplete penetrance for atypical and typical 22q11.2 deletions, and thus the increased difficulty in selecting patients for molecular testing. Our findings contribute to the genotype–phenotype data in different types of 22q11.2 deletions.

Although chromosomal microarray is the standard in the genetic diagnosis of 22q11.2DS, the method is relatively expensive and laborious. Our study proposes the use of the MLPA technique with combined kits for the diagnosis of these patients: first a screening MLPA for all suspected cases, followed by a follow-up MLPA test to characterize the size and exact location of the deletion. The results obtained have the advantage of being a faster, simpler, and cheaper alternative—very important characteristics, especially for a low-income region.

## Figures and Tables

**Figure 1 genes-13-02083-f001:**
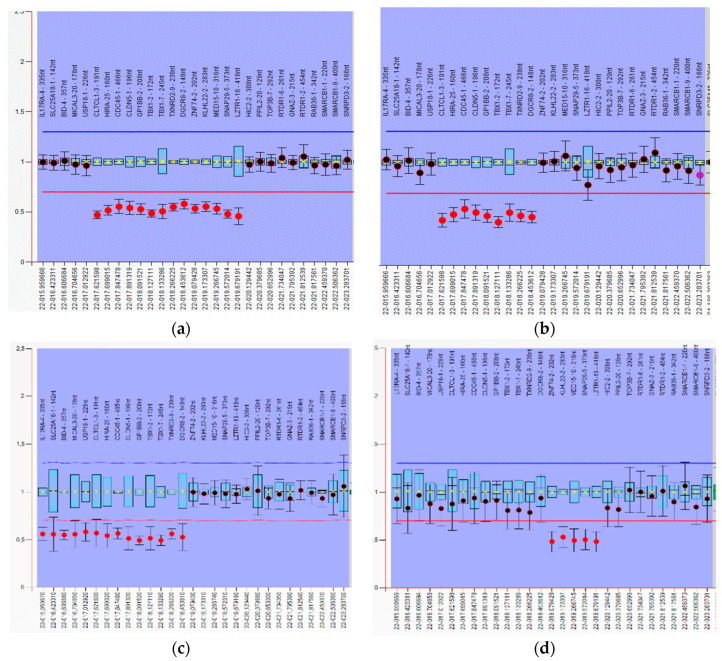
Partial images of atypical deletion in 22q11.2 region identified by MLPA P250 kit (generated by Coffalyser software, in bar chart format, red spots—ratio < 0.7) (**a**) heterozygous LCR A–D deletion (case 23); (**b**) heterozygous LCR A–B deletion (case 2) (**c**) heterozygous CES- LCR B deletion (case 1); (**d**) heterozygous LCR B-D deletion (case 9).

**Figure 2 genes-13-02083-f002:**
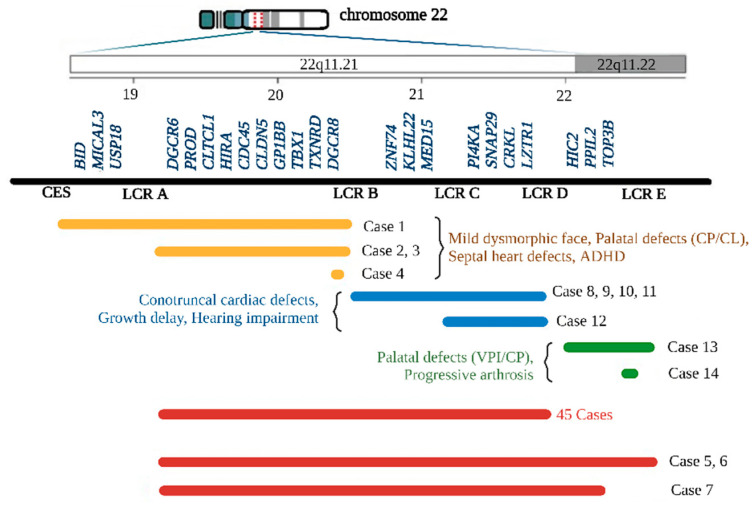
Different types of deletion and their frequent associated clinical signs identified by our study. With yellow bars: proximal deletion (cases 1–4), with blue bars: central deletion (cases 8–12), with green bars: terminal deletion (cases 13, 14) and with red bars: classical deletion. CP = Cleft palate, CL = Cleft lip, VPI = Velopharyngeal incompetence.

**Table 1 genes-13-02083-t001:** Phenotypic findings in LCR22 A–D deletions.

	Phenotypic Features	No. of Cases	F	M
Dysmorphism	Characteristic	32	14	18
	Mild	7	2	5
	Atypic	1	-	1
Heart defects	Tetralogy of Fallot	20	6	14
	Ventricular septal defect	15	5	10
	Truncus arteriosus	4	1	3
	Aortic arch anomalies	3	-	3
	Pulmonary atresia	3	-	3
	Atrial septal defect	2	-	2
	Patent ductus arteriosus	1	-	1
	Pulmonary stenosis	1	1	-
	Interrupted aortic arch	1	-	1
	Bicuspid aortic valve	1	-	1
	Other	4	2	2
Palatal defects	Velopharyngeal incompetence	20	11	9
	Dysphagia	8	2	6
	Cleft Palate	5	1	4
	Submucosal cleft palate	2	-	2
	Cleft lip and palate	1	1	-
Development delay ± cognitive impairment	Mild	14	6	8
	Moderate	8	3	5
	Severe	5	2	3
Psychiatric/Behavior problems	Attention deficit	7	5	2
	Autism-spectrum disorders	2	1	1
	Schizophrenia	1	1	-
Hearing impairment		5	3	2
Growth delay		18	5	13
Hypotonia		2	-	2
Hypocalcemia		21	10	11
Seizures		16	9	7
Immunodeficiency		21	10	11
Urogenital anomaly	Hydronephrosis,	7	5	2
	Cryptorchidism	3	-	3
	Renal hypoplasia	2	1	1
	Multicystic dysplastic kidney	1	1	-
	Hypospadias	1	-	1

**Table 2 genes-13-02083-t002:** Phenotypic findings in atypical deletions.

Case No.	1	2	3	4	5	6	7	8	9	10	11	12	13	14
Deletion type	CES to B	A to B	A to B	*DGCR8*	A to E	A to E	A to D+	B to D	B to D	B to D	B to D	C to D	D to E	*TOP3B*
Sex	M	M	F	M	F	M	F	F	F	M	M	F	F	M
Age at last evaluation	7 y	11 y	12 y	12 y	8 y	1 y 3 mo	1 y 8 mo	27 y	8 y	3 y	7 y	14 y	16 y	6 y
Inheritance	De novo	De novo	?	De novo	De novo	?	De novo	?	Maternal	?	De novo	De novo	De novo	De novo
Dysmorphism	Mild	Atypic	+	Atypic	+	+	+	-	+	+	+	+	-	-
Microcephaly	-	+	-	-	-	Macrocephaly	+	-	-	-	+	-	-	-
Heart defects	VSD, ASD	VSD, CoA	VSD, ASD	ASD	-	-	TOF	-	TOF	TOF	VSD ASD	VSD	-	-
Palatal defects	CL, CP	VPI	-	CP	-	VPI	CP	-	-	-	-	-	VPI	CP
Development delay ± cognitive impairment	-	-	Mild	Moderate	Moderate	Severe	Moderate	-	-	-	Mild	-	-	-
Psychiatric/Behavior	-	ADHD	-	ADHD	ADHD	-	-	-	-	-	-	-	-	-
Hearing impairment	-	-	-	+	+	±	±	-	-	-	-	-	-	-
Growth delay	-	-	-	-	-	+	-	-	-	-	+	+	-	-
Hypotonia	-	-	-	+	-	-	-	-	-	-	-	-	-	-
Hypocalcemia	-	+	+	-	-	+	-	-	-	-	-	-	-	-
Seizures	-	+	+	-	-	+	-	-	±	-	-	-	+	-
Immunodeficiency	-	-	-	-	+	+	+	-	-	-	-	-	-	-
Urogenital anomaly	-	-	-	RA, hypospadias	RA	HN	-	-	HN, RA	-	-	-	-	-
Other features	-	-	-	Auditory canal hypoplasia	Spastic tetraplegia	Cerebral atrophy, polydactyly	Spina bifida, polydactyly	-	-	-	CaL	CaL, hemangiomas	Progressive arthrosis	-

+ = present; - = not present; y = years; mo. = months; ? = Unknown; VSD = ventricular septal defect; ASD = atrial septal defects; TOF = Tetralogy of Fallot; CoA = Coarctation of the Aorta; VPI = Velopharyngeal incompetence; R = right; L = left; CP = Cleft palate; CL = Cleft lip; RA = Renal agenesis; HN = Hydronephrosis; CaL = Café au lait spots.

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
