# Peer review of "Different Types of Deletions Created by Low-Copy Repeats Sequences Location in 22q11.2 Deletion Syndrome: Genotype–Phenotype Correlation"

_genes, 2022, doi:10.3390/genes13112083_

Round 1

Reviewer 1 Report

In this paper, authors present the results of retrospective analysis of 59  cases with deletion in 22q11.2 region. Since this is the most common microdeletion syndrome there have been multiple studies on much larger cohorts in the literature, including meta-analysis on thousands of cases. The authors claim that they performed genotype-phenotype correlation.  However, in the current version of the manuscript authors do not attempt to perform more in-depth analysis of such correlations. They present the observed phenotypes, their prevalence in the cohort, but no systematic analysis (besides Table 2) of their association with different deletion regions was shown.  In Discussion, authors compare their findings to the data from the literature. They suggest that their results may differ from results obtained in other studies (e.g. "In our patients, a slightly higher prevalence of cardiac defects (71.1%) and 148 especially the occurrence of Fallot tetralogy was observed (45.2%)"). Authors do not check if these differences are statistically significant and do not discuss what could be the cause of such differences. To summarize, I believe that this study could provide some valuable insight for understanding the phenotypic spectrum of 22q11.2 deletion syndrome, however the manuscript and data presentation should be reorganized to focus on the novel findings that could enrich the current knowledge.   Minor comments: 1. I do not think Figure 1 provides any important information. This kind of figure can be placed in supplementary data.  2. Data in Table 1 and Table 2 are redundant. I would suggest squeezing these two into one table. 

Author Response

Dear reviewer,

Thank you for appreciating the manuscript, for the suggestions and for the concise evaluation. Please find your suggestions and our answers below:

In this paper, authors present the results of retrospective analysis of 59  cases with deletion in 22q11.2 region. Since this is the most common microdeletion syndrome there have been multiple studies on much larger cohorts in the literature, including meta-analysis on thousands of cases. The authors claim that they performed genotype-phenotype correlation.  However, in the current version of the manuscript authors do not attempt to perform more in-depth analysis of such correlations. They present the observed phenotypes, their prevalence in the cohort, but no systematic analysis (besides Table 2) of their association with different deletion regions was shown.  In Discussion, authors compare their findings to the data from the literature. They suggest that their results may differ from results obtained in other studies (e.g. "In our patients, a slightly higher prevalence of cardiac defects (71.1%) and 148 especially the occurrence of Fallot tetralogy was observed (45.2%)"). Authors do not check if these differences are statistically significant and do not discuss what could be the cause of such differences. To summarize, I believe that this study could provide some valuable insight for understanding the phenotypic spectrum of 22q11.2 deletion syndrome, however the manuscript and data presentation should be reorganized to focus on the novel findings that could enrich the current knowledge.   Minor comments: 1. I do not think Figure 1 provides any important information. This kind of figure can be placed in supplementary data.  2. Data in Table 1 and Table 2 are redundant. I would suggest squeezing these two into one table. 

We appreciate your suggestions, and we added the correction with statistical analysis in the results part, lines 128-141. We have described a potential bias that could account for our findings. We also updated the literature review and reorganized the structure of the manuscript.

We believe that figure 1 highlights the value of the MLPA diagnosis of 22q11.2DS as a simple and thorough technique.

We have split the results in 2 tables because table 1 describes typical deletions and table 2 atypical cases, squeezing them in one table will make the data difficult to understand. We have simplified the tables and hope that now they are less redundant.

Thank you for your valuable suggestion that improved the quality of our manuscript!

Reviewer 2 Report

In this manuscript, the authors reported the molecular analyses of 59 cases with 22q11.2 deletion syndrome using MLPA to define the extent of the genomic deletion. The majority of them present the classic deletion whereas 14 patients carry atypical deletions in the region. A retrospective study aiming at assessing a genotype-phenotype correlation is reported.

I believe that the study has the potential to give important information on the implication of the different genes or combination of them within the 22q11.2 region but at the same time the data should be further elaborated and more clearly presented.

Major point:

1) The genotype-phenotype correlation claimed in the title is not evident in the manuscript. This issue is addressed in the Discussion but as it stands now it is difficult to follow. I think that this analysis should be part of the results and should be also graphically represented in a figure (e.g. a sort of Figure 2 showing the deleted regions together with the associated clinical signs).

2) The Figures should be provided at higher resolution, even enlarging it is very difficult to read them.

3) The legend of Figure 1 and Figure 2 should be more explicative of what is shown in the figure.

4) Table 1 summarising the phenotypic features in patients carrying LCR A-D deletions should be integrated with the information of sex relative to each clinical sign.

5) The 3 cases reported in line 90/91 with LCR A-E do include the 2 cases with A-E deletion and the 1 with the A-beyond D mentioned in Table 2? This should be better reported.

Author Response

Dear reviewer,

Thank you for all your comments. They were really useful and definitely contributed to the improvement of the manuscript. We shall take them in order:

In this manuscript, the authors reported the molecular analyses of 59 cases with 22q11.2 deletion syndrome using MLPA to define the extent of the genomic deletion. The majority of them present the classic deletion whereas 14 patients carry atypical deletions in the region. A retrospective study aiming at assessing a genotype-phenotype correlation is reported.

I believe that the study has the potential to give important information on the implication of the different genes or combination of them within the 22q11.2 region but at the same time the data should be further elaborated and more clearly presented.

Major point:

 1) The genotype-phenotype correlation claimed in the title is not evident in the manuscript. This issue is addressed in the Discussion but as it stands now it is difficult to follow. I think that this analysis should be part of the results and should be also graphically represented in a figure (e.g. a sort of Figure 2 showing the deleted regions together with the associated clinical signs).

 We have restructured the manuscript according to your advice and updated the results. We have upgraded the figure 2 with information about the phenotype.

2) The Figures should be provided at higher resolution, even enlarging it is very difficult to read them.

 Thank you for your kind observations, we have substituted the figures with ones with a higher resolution.

3) The legend of Figure 1 and Figure 2 should be more explicative of what is shown in the figure.

We have included explanatory data for the figures. Thank you for asking this!

4) Table 1 summarising the phenotypic features in patients carrying LCR A-D deletions should be integrated with the information of sex relative to each clinical sign.

We have included changes according to these valuable indications.

5) The 3 cases reported in line 90/91 with LCR A-E do include the 2 cases with A-E deletion and the 1 with the A-beyond D mentioned in Table 2? This should be better reported.

Yes, all these three cases included the 2 cases with A-E, and one with A beyond D. We corrected this information in lines 87-91.

Reviewer 3 Report

In this manuscript, the authors demonstrated different types of deletions in sequence location in 22q11.2 syndrome and genotypic and phenotypic correlation. This study is helpful to understand the 22q11.2 deletion syndrome having novel deletions and disease phenotypes and advances the way to understand the patient genetic abnormality using the MLPA technique. I have the following recommendations to improve this manuscript.

Concerns

1. Table 2 is difficult to understand, could you please split it into two tables?

2. This study has limitations in understanding the disease, please highlight them in the discussion section

3. Need to provide the images that represents ASD/VSD versus phenotype data

Author Response

Thank you for appreciating the manuscript and for suggestions that improved the quality of the manuscript. Please find your suggestions and our answers below:

  1. Table 2 is difficult to understand, could you please split it into two tables?

We have simplified the table 2 and e think that now it is easier to understand

  1. This study has limitations in understanding the disease, please highlight them in the discussion section

Indeed, we agree with your suggestions and illustrated the potential bias in the discussion section, in the lines 195-207.

  1. Need to provide the images that represents ASD/VSD versus phenotype data

We have updated the figure 2 with clinical aspects. We anticipate that this clears up the phenotypic image.

Round 2

Reviewer 2 Report

The authors properly addresses the raised issues. I would just recommend to:

1. Check for typos

2. Be sure that Figure 2 in the final version is complete (at present it appears a bit cropped on the right)